

# Vertical farming for lettuce production in limited space: a case study in Northern Thailand

Suwimon Wicharuck[1,2], Nuttapon Khongdee[3], Ar Man[4],
Wahyu Nurkholis Hadi Syahputra[5], Parichat Yalangkan[2],
Prapaporn Chaiphak[2] and Chatchawan Chaichana[2]

[1] Office of Research Administration, Chiang Mai University, Chiang Mai, Thailand
[2] Energy Technology for Environment Research Center, Department of Mechanical Engineering, Faculty of Engineering, Chiang Mai University, Chiang Mai, Thailand
[3] Department of Highland Agriculture and Natural Resources, Faculty of Agriculture, Chiang Mai University, Chiang Mai, Thailand
[4] Graduate Master's Degree Program in Energy Engineering, Department of Mechanical Engineering, Faculty of Engineering, Chiang Mai University, Chiang Mai, Thailand
[5] Agricultural Engineering Program, Department of Mechanical Engineering, Faculty of Engineering, Chiang Mai University, Chiang Mai, Thailand

Corresponding author
Chatchawan Chaichana,
c.chaichana@eng.cmu.ac.th

## ABSTRACT

**Background:** Greenhouse vertical farming under natural sunlight is an alternative farming technique that grows crops in a stacking column and extends in a vertical direction. Sunlight availability is one of the crucial factors for crop development in vertical farming. Therefore, this investigation aimed to examine the effect of sunlight availability on lettuce growth and yields at different levels of vertical shelves.

**Methods:** Six shelves were constructed with three levels: upper, middle and lower levels. Lettuces (*Lactuca sativa* L.) as 'Baby Cos' and 'Green Oak' at 14 days after sowing were planted on the three levels. The photosynthetic photon flux density (PPFD) was recorded, and the PPFD values were then converted to the daily light integral (DLI). Plant height and canopy width were measured three times at 14, 21 and 28 days after transplanting. At maturity, fresh weight (FW) was directly monitored after harvest.

**Results:** The results showed that the highest PPFD and DLI values were found at the upper level (PPFD 697 $\mu$mol m$^{-2}$ s$^{-1}$ and DLI 29 mol m$^{-2}$ d$^{-1}$) in comparison to the middle (PPFD 391 $\mu$mol m$^{-2}$ s$^{-1}$ and DLI 16 mol m$^{-2}$ d$^{-1}$) and lower (PPFD 322 $\mu$mol m$^{-2}$ s$^{-1}$ and DLI 13 mol m$^{-2}$ d$^{-1}$) levels. The lowest plant height and canopy width values were observed on the upper levels for both lettuce varieties during the three measurement dates. The middle ('Baby Cos' = 123.8 g plant$^{-1}$ and 'Green Oak' = 190.7 g plant$^{-1}$) and lower ('Baby Cos' = 92.9 g plant$^{-1}$ and 'Green Oak' = 203.7 g plant$^{-1}$) levels had the higher values of FW in comparison to the upper level ('Baby Cos' = 84.5 g plant$^{-1}$ and 'Green Oak' = 97.3 g plant$^{-1}$). The values of light use efficiency (LUE) showed an increased trend from the upper to lower levels in both varieties, with values of 'Baby Cos' of 0.10 g mol$^{-1}$ in the upper level, 0.28 g mol$^{-1}$ in the middle level and 0.26 g mol$^{-1}$ in the lower level and 'Green Oak' of 0.12 g mol$^{-1}$ in the upper level, 0.44 g mol$^{-1}$ in the middle level and 0.57 g mol$^{-1}$ in the lower level. The findings of the study indicated the viability of utilizing vertical shelves for lettuce production.

# INTRODUCTION

During the last few decades, climate change has caused difficulties in open-field crop production (*Ikonomopoulos & Tsilingiridis, 2016*), and it has directly affected crop quality and yields. Agriculture in urban areas has gained popularity, especially in indoor areas, such as on rooftops, facades, and balconies. Numerous novel farming practices and technologies have been created to enhance crop yield per unit area within a limited farmland, such as vertical farming (VF). The VF is an innovative farming technique that grows crops in a stacking column and extends vertically. The VF can increase planting density (the number of plants per unit area) compared to horizontal or conventional farming (*Touliatos, Dodd & Mcainsh, 2016*). This innovative technique optimizes space utilization and resource efficiency, which is particularly suitable for regions with limited arable land. The VF also offers advantages in year-round cultivation, reduced water usage, and minimal pesticide requirements.

Greenhouse VF cultivation under natural sunlight is another possibility for crop production in a limited space. Crops can directly utilize sunlight as a source for the photosynthesis process without the supplementation of a lighting system. The photosynthetic process involves the conversion of sunlight into chemical energy and the transformation of $CO_2$ and $H_2O$ into glucose (*Arnon, 1959*). The characteristics of sunlight in terms of quality and quantity are essential for crop development in VF. Plants benefit from sunlight with wavelengths ranging from 400 to 700 nm. This wavelength range is considered photosynthetically active radiation (PAR), the light spectrum for the plant photosynthetic process. The number of photons received by plants from PAR light can be measured per unit area as photosynthetic photon flux density (PPFD, $\mu mol\ m^{-2}\ s^{-1}$) or daily light integral (DLI, $mol\ m^{-2}\ d^{-1}$). Each crop needs a specific amount of sunlight for crop development; for example, DLI values were 8–14 $mol\ m^{-2}\ d^{-1}$ for lettuce (*Baumbauer, Schmidt & Burgess, 2019*), 30–35 $mol\ m^{-2}\ d^{-1}$ for tomato (*Spaargaren, 2001*) and 14–45 $mol\ m^{-2}\ d^{-1}$ for strawberry (*Bosa et al., 2019*).

The characteristic of sunlight availability on the VF decreased from the upper to the lower parts and between each level of vertical shelves because of shading. The shade from neighboring shelves can diminish the quantity of incident sunlight reaching the plants between shelves and levels. Additional cultivating space must also be provided in vertical directions. Therefore, the design of vertical shelves must be taken into account when the concept of the VF is implemented in terms of these three topics: (i) distance between shelves, (ii) space from upper to lower levels and (iii) orientation of shelves. However, the VF has shown some promise in raising crop yields and addressing difficulties related to a lack of land; research on lettuce production in locations with limited space, particularly in Northern Thailand, remains restricted.

Lettuce is one of Thailand's most important vegetable crops, contributing significantly to domestic consumption and export activities, with a fresh and chilled export volume of 2,772 t in 2021 (*WITS, 2023*). Cultivation practices are typically conducted year-round

using traditional and modern farming techniques, such as open-field and protected cultivation (greenhouse or closed chamber). Therefore, lettuce, a widely consumed leafy green vegetable, holds particular promise for vertical farming due to its compact growth habit and relatively short cultivation period. Through the implementation of vertical farming methods, lettuce production can be significantly enhanced by utilizing vertical space efficiently and allowing numerous layers of crops to be cultivated at the same time.

In our previous research (*Wicharuck et al., 2023*), lettuce growth and yield between horizontal shelves (HS) and vertical shelves (*VS*) under natural sunlight in Chiang Mai province were compared. The results indicated that light was a limiting factor for lettuce growth. HS showed higher fresh and dry biomass, but *VS* had 1.5 times the plant density of HS. In a different study conducted by *Chaichana et al. (2022)*, a computational model, the RHINO with the Grasshopper plug-in, was used for estimating the annual availability of sunshine on vertical shelves in Chiang Mai province, Thailand. Under this investigation, the size of the shelves remained the same while different shelf orientations and seasons were examined. The results from the model indicated that (i) the daily PPFD and DLI were higher from the lower to higher levels, with the values of DLI ranging from 5 to 36 mol $m^{-2}$ $d^{-1}$, approximately, and (ii) the north-south-oriented shelf provided more homogeneous PPFD and DLI than the east–west orientation. The findings from those two studies can be used to select specific crops that correspond to the amount of sunlight available at a given location. The research gap in this study lies in the need for comprehensive studies that establish a connection between the observed variations in sunlight availability on vertical shelves and crop-specific responses and growth patterns. Further study is needed in order to test this hypothesis. Therefore, this investigation aimed to address the existing research gap and make a valuable contribution to the current knowledge base by examining the potentiality of vertical farming for lettuce production in limited-space areas of Northern Thailand. The research evaluated the productivity and economic viability of implementing lettuce-specific vertical farming systems by considering crop growth, yields, resource consumption, and income generation factors. This study also provided comprehensive insights into the advantages and feasibility of vertical farming for lettuce production in the region.

## MATERIALS AND METHODS

### Experimental setup

The study was conducted in Hang Dong district, Chiang Mai province, Thailand, in February 2022 (Fig. 1A). The average temperature was 27.5 ± 8.3 °C, and the relative humidity was 62.4 ± 27.6%, with the average highest value of solar radiation during the daytime being 950 w $m^{-2}$ (Fig. 1B). Six rectangular shelves were constructed with dimensions of 2.2 m in height, 0.3 m in width and 7.5 m in length. The shelves were set 0.8 m apart inside the greenhouse, with a width of 10 m, a length of 10 m, and a height of 2.5 m. Each shelf was consisted of three different levels (upper, middle and lower levels), and the distance between levels was 0.6 m. Each planting tray was prepared with 0.15 mm thick of transparent plastic (Figs. 1C and 1D ). The PPFD (µmol $m^{-2}$ $s^{-1}$) was recorded using WatchDog LightScout Quantum Light Sensors (PAR light sensors; Spectrum

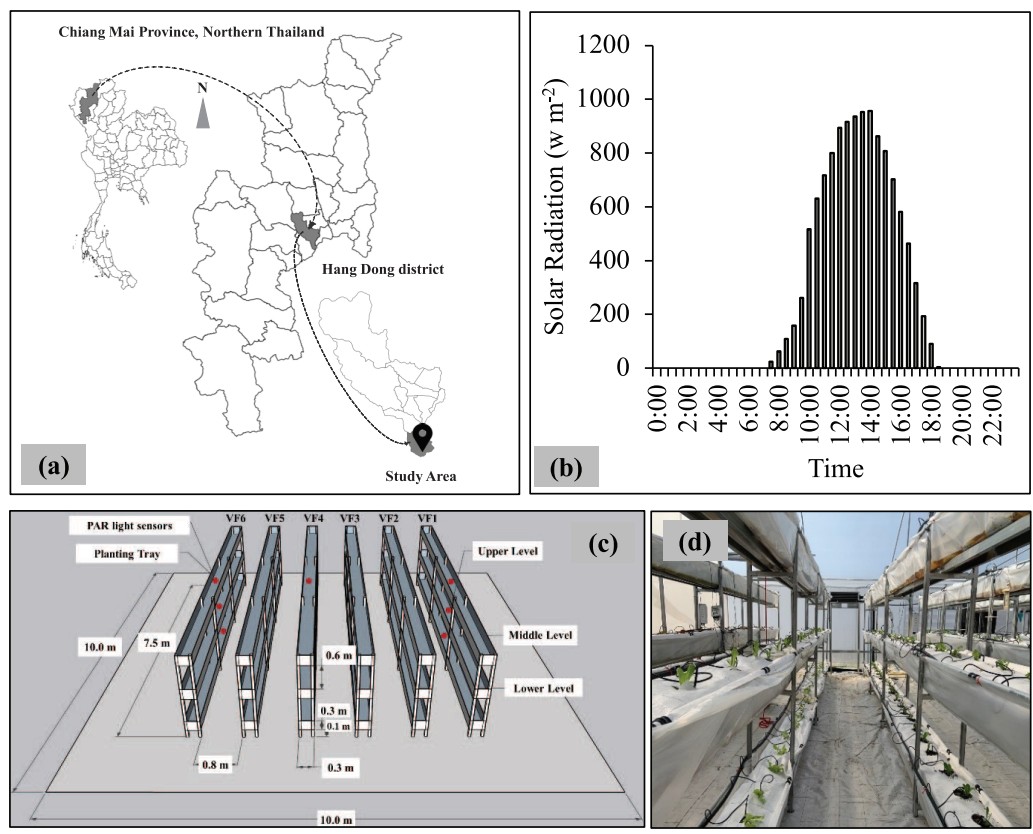

**Figure 1** (A) Location of the study area, administrative boundary data taken from *OCHA (2023)*. (B) Average values of solar radiation at the study area. (C) Layout of vertical shelves inside the greenhouse, and (D) Inside-view of experimental room.

Technologies, Inc., Aurora, Illinois, USA). Nine PAR light sensors were installed at a height of 0.2 m above the surface of growing media in the middle of three different levels of VS1, VS4 and VS6 (Fig. 1C).

## Preparation of growing media and lettuce cultivation

Growing media consisted of topsoil (from the Faculty of Agriculture, Chiang Mai University, Chiang Mai, Thailand): chicken manure: coconut husk: rice husk ash: rice husk in a ratio of 2:1:1:1:1 (by volume) with 50 g of rock phosphate and 50 g of charcoal. A total of 7 kg of growing media were added to a white growing bag, with the diameter and height of each bag being 0.3 m and 0.3 m, respectively. Then, the growing bags were placed inside the planting tray. The surface of each planting tray was covered by white mulching plastic. For lettuce cultivation, lettuce (*Lactuca sativa* L.) varieties of 'Baby Cos' and 'Green Oak' planted in coconut coir were purchased from a local nursery after 14 days of sowing and transplanted into white growing bags. The same variety of lettuce was planted at each level. The total number of lettuces per level was 25 plants, and the total per shelf was 75 plants (25 plants × 3 levels). There were 225 lettuces per variety. A drip irrigation system was installed for each level and each growing bag (Fig. 1D). The irrigation water amounts were calculated using the Penman-Monteith equation (*Allen et al., 1998*), and the irrigation

water was automatically distributed 2 min time$^{-1}$ twice daily. The total amount of water was approximately 400 cm$^3$ plant$^{-1}$ d$^{-1}$.

## Data collection

Six levels were selected for data collection: three different levels (up, mid and low) for 'Baby Cos' and three levels (up, mid, and low) for 'Green Oak'. The experiment consisted of three treatments with three replications per level: (i) the upper level, (ii) the middle level and (iii) the lower level. The upper level was set as the control treatment due to the lack effect of shading from the other shelves and levels. Different parameters were evaluated, such as plant growth (height and canopy width), fresh and dry weight and light use efficiency. Plant height and canopy width were measured by a vernier calliper at 14, 21 and 28 days after transplanting (DAT) for both 'Green Oak' and 'Baby Cos'. At maturity (28 DAT), the fresh weight (FW) of 'Green Oak' and 'Baby Cos' was weighted directly after harvesting; then, the plant samples were dried for a duration of three days at temperature of 65 °C, and the dry weight (DW) was evaluated. The PPFD (μmol m$^{-2}$ s$^{-1}$) was recorded every 30 min. Then, the values of PPFD were converted to the daily light integral (DLI, mol m$^{-2}$ d$^{-1}$) and light use efficiency (LUE, g mol$^{-1}$) for each level, as shown in Eqs. (1) and (2). The light hour (LH) was counted between the first and last hour of the day that the PAR light sensors recorded the PPFD values. In addition, the LUE was calculated as the FW divided by the cumulative DLI at 28 DAT.

$$DLI = PPFD \times LH \times 0.0036 \tag{1}$$

$$LUE = FW \div \text{Cumulative DLI at 28 DAT.} \tag{2}$$

## Statistical analysis

The statistical analysis was calculated for each lettuce type at three different levels using Sigma Plot 15 software. A One-way ANOVA was used to compare differences of means in plant growth (height and canopy width) and yields (fresh and dry weight) as well as LUE for three levels on each lettuce variety, followed by a *post-hoc* test of Turkey's honestly significant difference. Pearson's correlations and linear regressions were calculated for (i) plant height and cumulative DLI and (ii) canopy width and cumulative DLI.

## RESULTS

### Variations of PPFD and DLI

Figure 2 shows the results of average PPFD at three different levels and daily PPFD and DLI during the measurement periods. The results can be explained as follows:

- The measured PPFD for each level showed a distinct pattern corresponding to the apparent motion of the sun over the sphere of the sky, with sunlight hours around 10.5–12.5 h d$^{-1}$. The values of average PPFD were lower from the upper to lower levels, and the highest peaks of PPFD values for each level were between 10:30 a.m. and 15:30 p.m. (Figs. 2A–2C). The average PPFD values during the highest peak of different
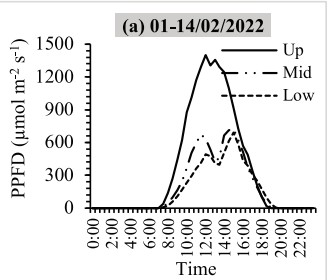
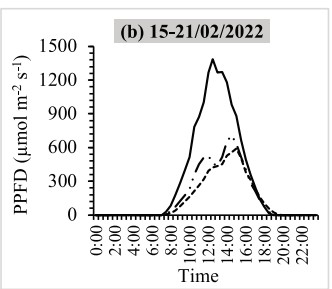
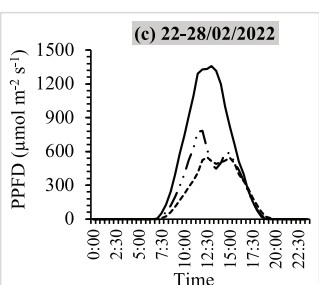

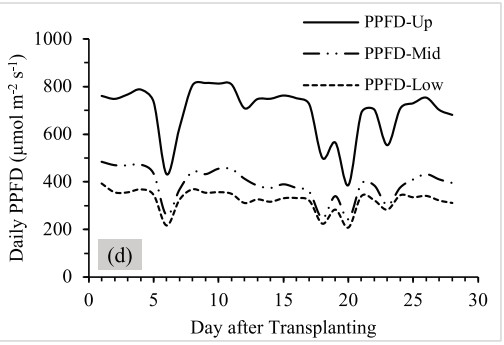
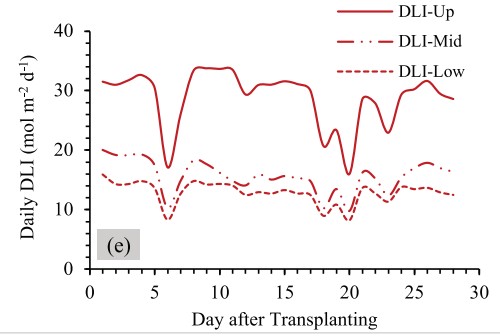

**Figure 2** (A–C) Average values of PPFD at three levels (upper-middle-lower), (D and E) daily PPFD and DLI during the measurement periods. Up, mid and low referred to upper, middle and lower levels.

levels were 514–1,385 μmol m$^{-2}$ s$^{-1}$ at the upper level, 231–621 μmol m$^{-2}$ s$^{-1}$ at the middle level and 161–581 μmol m$^{-2}$ s$^{-1}$ at the lower level.

- Daily PPFD and DLI variations displayed similar trends during the study periods (Figs. 2D and 2E). The upper level gave the highest average PPFD and DLI values, while the lowest average PPFD and DLI values were observed at the lower level.

- On average values of PPFD, the PPFD values were 697 ± 35 μmol m$^{-2}$ s$^{-1}$ at the upper level, 391 ± 19 μmol m$^{-2}$ s$^{-1}$ at the middle level and 322 ± 39 μmol m$^{-2}$ s$^{-1}$ at the lower level. The middle and lower levels received 43.9% and 53.8% less sunlight compared to the upper level, respectively (Fig. 2D).

- The average DLI values ranged from 17 to 34 mol m$^{-2}$ d$^{-1}$ at the upper level, 10 to 20 mol m$^{-2}$ d$^{-1}$ at the middle level and 8 to 16 mol m$^{-2}$ d$^{-1}$ at the lower level. Similar to the PPFD measurements, it was observed that the lower level (55.4%) and the middle level (45.7%) had less sunlight than the upper level. The cumulative DLI values in one crop cycle were 807 ± 39 mol m$^{-2}$ at the upper part, 438 ± 41 mol m$^{-2}$ at the middle part and 360 ± 52 mol m$^{-2}$ at the lower part (Fig. 2E).

## Plant growth and relationships of plant growth and cumulative DLI

Average plant height and canopy width values of 'Baby Cos' and 'Green Oak' during the measurement periods are shown in Fig. 3. For both varieties, the plant height and canopy width values increased with the increasing time. Significant differences in plant height and canopy width on different levels were observed for 'Baby Cos' during three measurement

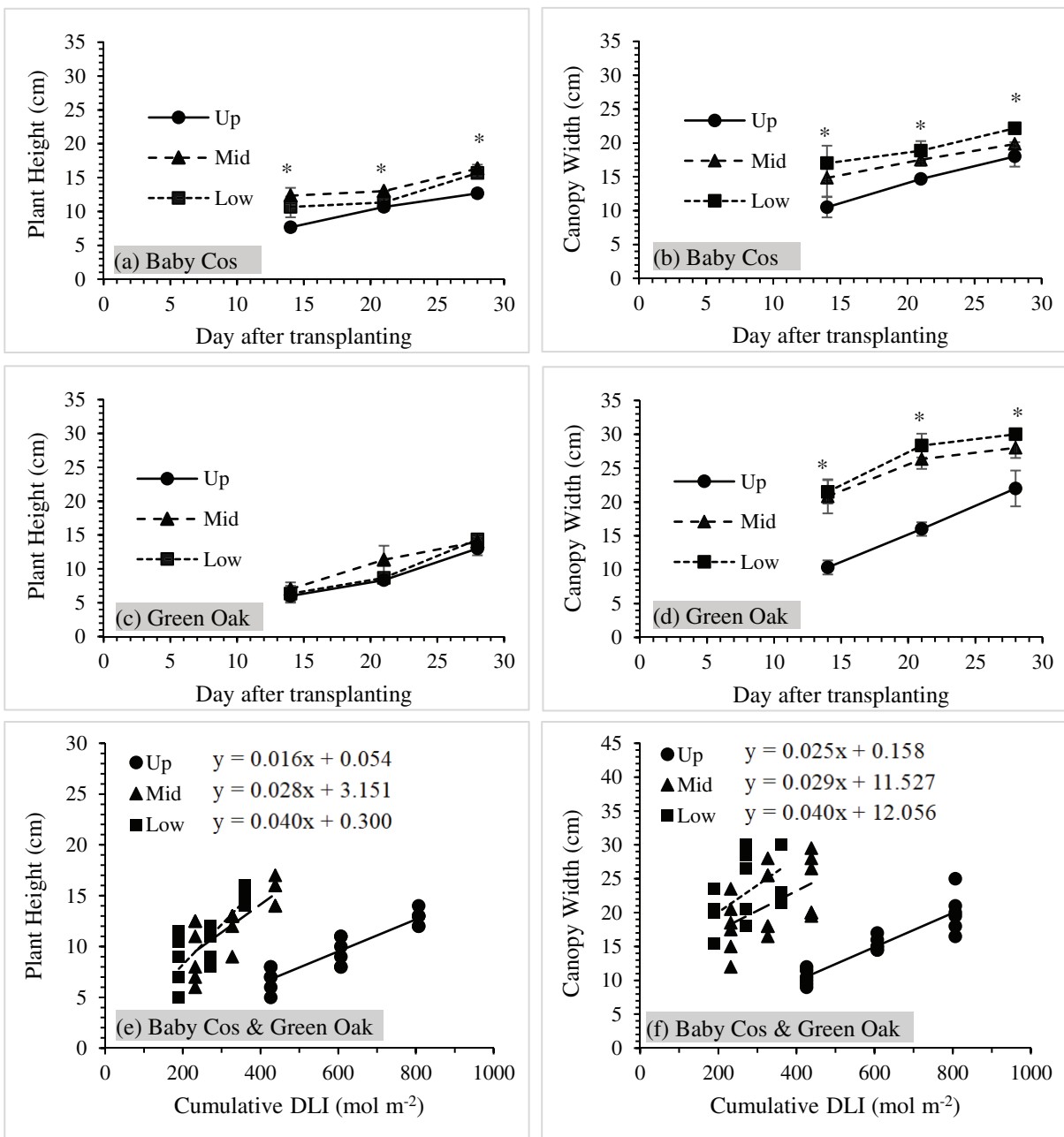

**Figure 3 (A–D) Average values of plant height and canopy width of 'Baby Cos' and 'Green Oak' at three measurement dates (14, 21 and 28 DAT). (E–F) relationships between plant growth (height and canopy width) and cumulative DLI of both 'Baby Cos' and 'Green Oak'.** Data points and error bars represented the mean and standard deviation. The asterisk (*) indicated significant differences in means at $p < 0.05$. Up, Mid and Low referred to Upper, Middle and Lower levels, respectively.

periods (Figs. 3A and 3B ). 'Baby Cos' at the upper level had the lowest plant height and canopy width values, and the highest plant height and canopy width values were found at the middle and lower levels, respectively.

For 'Green Oak', non-significant differences in plant height were observed at three levels during the study periods (Fig. 3C). The middle level tended to give the highest values

**Table 1** Mean values and standard deviation (SD) of fresh and dry yields as well as LUE of BC ('Baby Cos') and GO ('Green Oak') at 28 DAT.

| Parameters | | Unit | Up | Mid | Low | Average | P-values |
|---|---|---|---|---|---|---|---|
| Fresh yields | BC | g plant$^{-1}$ | 84.5 ± 4.0b | 123.8 ± 8.5a | 92.9 ± 12.1b | 100.4 ± 20.7 | 0.004* |
| | GO | | 97.3 ± 16.3b | 190.7 ± 14.7a | 203.7 ± 30.9a | 163.9 ± 58.0 | 0.002* |
| Dry yields | BC | g plant$^{-1}$ | 4.56 ± 0.24 | 4.63 ± 0.87 | 4.25 ± 0.72 | 4.48 ± 0.20 | NS |
| | GO | | 8.83 ± 1.87 | 9.43 ± 0.93 | 9.46 ± 1.50 | 9.24 ± 0.36 | NS |
| Cumulative DLI | | mol m$^{-2}$ | 807 | 438 | 360 | – | – |
| LUE | BC | g mol$^{-1}$ | 0.10 ± 0.01b | 0.28 ± 0.02a | 0.26 ± 0.03a | 0.22 ± 0.10 | <0.001* |
| | GO | | 0.12 ± 0.02c | 0.44 ± 0.03b | 0.57 ± 0.09a | 0.37 ± 0.23 | <0.001* |

**Note:**
Up, Mid and Low referred to the upper, middle and lower levels. * and NS were significant and non-significant differences in means at $p < 0.05$. Different letters indicated statistically significant differences in means at $p < 0.05$ as determined by the Turkey HSD's test.

compared to the upper and lower levels. Furthermore, the average values of canopy width at three levels were significantly different under all three measurements (Fig. 3D). The lowest values of canopy width occurred at the upper levels compared to the others.

Moreover, Figs. 3E and 3F display the relationships between plant growth (height and canopy width) and the cumulative DLI of 'Baby Cos' and 'Green Oak' at three different levels during the study periods. Significant positive correlations between plant growth and the cumulative DLI were observed for both lettuce varieties at $P < 0.05$. For plant height *vs.* the cumulative DLI, the determination of coefficient ($R^2$) values was 0.849, 0.606 and 0.715 for the upper, middle and lower levels, respectively. The $R^2$ values between canopy width and the cumulative DLI were 0.834 at the upper level, 0.256 at the middle level and 0.320 at the lower level. These values pointed out that plant growth in height and canopy width tended to increase with light intensity at three different levels.

## Yields and light use efficiency

Table 1 shows the fresh (FW) and dry (DW) weight results for both 'Baby Cos' and 'Green Oak' at the 28 DAT. Significant differences were observed in the average FW for both lettuce varieties. The lowest FW values for 'Baby Cos' (84.5 g plant$^{-1}$) and 'Green Oak' (97.3 g plant$^{-1}$) were found in the upper level when compared to the middle and lower levels. This suggests that lettuce plants grown at the upper level experienced a reduction in biomass accumulation compared to the other levels. In addition, average values of DW were not significantly different at three levels for both lettuce varieties.

For light use efficiency (LUE), both 'Baby Cos' and 'Green Oak' had significantly lower values of LUE at the upper level in comparison to the middle and lower levels, with LUE values of 0.10 g mol$^{-1}$ in 'Baby Cos' and 0.12 g mol$^{-1}$ in 'Green Oak' (Table 1). This indicates that lettuce plants grown at the upper level had lower efficiency in converting light energy into biomass than those at the other levels.

## DISCUSSION

Sunlight availability in vertical farming is a primary factor affecting plant growth (height and canopy width) and, finally, crop development and yields. This is because plants convert photons from light into energy through the photosynthesis process. Therefore,
each crop requires a particular light intensity for optimal crop growth and productivity (*Wijaya, Sigmarawan & Budisanjaya, 2019*; *Sutulienė et al., 2022*; *Zhou, Li & Wang, 2022*).

The findings of this study showed the importance of considering the availability of sunlight when designing and managing vertical farming systems (Fig. 2). Different levels of PPFD and DLI emphasized the need for strategic crop positioning to ensure optimal light exposure for growth and productivity. Adjusting the shelf design and cultivation techniques can help maximize light utilization and promote uniform crop development throughout the vertical farming system. In addition, the differences in plant height and canopy width may be attributed to variations in light exposure, air circulation, and nutrient availability at different levels (*Zhang & Kacira, 2022*). It is worth noting that the response to vertical positioning will differ among crop varieties (*Kang et al., 2013*; *Sutulienė et al., 2022*), as in the contrasting patterns between 'Baby Cos' and 'Green Oak' (Fig. 3).

From Fig. 2, the amounts of sunlight intensity (PPFD and DLI values) decreased from the upper to lower levels because of shading from nearby shelves and above levels (*Linsley-Noakes, Wilken & de Villiers, 2004*; *Wicharuck et al., 2023*). This indicated that the availability of sunlight was reduced from the upper to the lower levels, with the average values of PPFD (697 $\mu$mol m$^{-2}$ s$^{-1}$) and DLI (29 mol m$^{-2}$ d$^{-1}$) being more than twice as high on the upper level in comparison to the lower levels. Nevertheless, higher sunlight intensity did not always increase plant growth and yields. This was correlated to the studies of *Pennisi et al. (2020)* and *Zhou, Li & Wang (2022)*, who explained that increasing light intensity did not have a significant impact on crop yields. The average value of PPFD at the upper level was about more than 1,000 $\mu$mol m$^{-2}$ s$^{-1}$ during the daytime, but Fig. 3 shows that the upper level exhibited the lowest plant growth and yield values compared to the middle and lower levels. This highlights the complexity of the relationship between light intensity and crop productivity. Every crop requires optimal saturation points to grow. For example, kale and spinach reached their maximum photosynthetic rates at light intensities of approximately 800 $\mu$mol m$^{-2}$ s$^{-1}$ and 1,200 $\mu$mol m$^{-2}$ s$^{-1}$, respectively (*Erwin & Gesick, 2017*). While plants require a certain level of light for photosynthesis and growth, excessive light intensity can lead to diminishing returns and may even have detrimental effects on plant development (*Roeber et al., 2021*). This implies that factors other than just the amount of light, such as shadowing, are responsible for controlling the growth and development of crops (*López-Marín et al., 2012*). Therefore, applying shading screens can help mitigate the thermal and light stress during a crop cycle spanning both winter and summer seasons for some crops (*López-Marín et al., 2012*).

The values of LUE for the middle and lower levels were more than 50% higher than the upper level. This is because crop development is related to the amount of sunlight reaching the canopy. For lettuce, the saturation point of light in terms of PPFD value is approximately 500 $\mu$mol m$^{-2}$ s$^{-1}$, according to the study of *Tsoumalakou et al. (2022)*. Increasing sunlight can enhance crop growth and yield if it does not surpass a critical threshold. Further increments in sunlight may not contribute to additional growth-promoting benefits, as crops exhibit optimal productivity within a specific range of light intensity (*Weiguo et al., 2012*). Therefore, optimizing light conditions within the

**Table 2 Comparison of average lettuce fresh weight values from different cultivation methods and locations.**

| Sources | Location | Cultivation methods | | Lighting Source | PPFD (μmol m$^{-2}$ s$^{-1}$) | DLI (mol m$^{-2}$ d$^{-1}$) | LH (h) | T (°C) | RH (%) | PD (plant m$^{-2}$) | FW (kg m$^{-2}$) |
|---|---|---|---|---|---|---|---|---|---|---|---|
| Own study | Thailand | GH | VF | Sunlight | 322–697 | 8–34 | 12 | 27.5 | 62 | 48 | 6.34 |
| Wicharuck et al. (2023) | Thailand | GH | VF | Sunlight | 147–217 | 5.9–10.4 | 11.5 | 22–30 | 66–85 | 24 | 1.50 |
| | | | HF | | 245 | 14.4 | | | | 16 | 1.99 |
| Matysiak, Kaniszewski & Mieszczakowska-Frąc (2023) | Poland | PF | HS | Artificial Light | – | – | – | 23 | 75 | 40 | 3.40 |
| Touliatos, Dodd & Mcainsh, 2016 | UK | PF | VF | Artificial Light | 134–491 | 7.7–28.8 | 16 | 16–18 | 60–80 | 1,000 | 95.00 |
| | | | HS | | 340–570 | 19.6–32.8 | | | | 50 | 6.90 |
| Wang et al. (2023) | China | GH | SBS | Sunlight | – | – | – | 18–22 | 65 | 18 | 2.91 |
| | China | GH | HS | Sunlight | – | – | | | | 30 | 6.81 |

Notes:
PPFD, DLI, LH, T, RH, PD and FW were photosynthetic photon flux density, daily light integral, lighting hour, room temperatures, room relative humidity, planting density and fresh weight.
GH presented greenhouse and PF represented plant factory.
VF, HF, HS and SBS were vertical farming, horizontal farming, hydroponic system and soil-based system, respectively.

**Table 3 A preliminary calculation of economic analysis of vertical farming in comparison to horizontal farming.**

| Parameters | Vertical farming | Horizontal farming |
|---|---|---|
| **Investment cost**[A] | 285 USD per shelve | 95 USD per shelve |
| **Irrigated water** | | |
| 400 cm$^3$ plant$^{-1}$ | 19,200 cm$^3$ plant$^{-1}$ | 6,400 cm$^3$ plant$^{-1}$ |
| **Total yields** | | |
| Planting density[B] 16 plants m$^{-2}$ | For three levels = 48 | For one level = 16 |
| Average fresh yields[C] 132 g plant$^{-1}$ | 6.34 kg m$^{-2}$ | 2.11 kg m$^{-2}$ |
| **Income** (1 USD = 35 THB) 1 kg = 80 THB[D] | 14.5 USD | 4.8 USD |

Notes:
[A] Financial analysis from this study.
[B] Spacing between plants were 0.25 m$^2$.
[C] Average values between 'Baby Cos' (100 g plant$^{-1}$) and 'Green Oak' (164 g plant$^{-1}$).
[D] Fresh yield prices of mixed lettuce in Thai market (Talaadthai, 2023).

vertical farming system to match the specific light requirements of crops is essential. Crop selection is a good strategy for planning crops according to the available intensity of sunlight at each level, but it is also important to consider the height of plants in the VF system. The design of the vertical shelves also plays a significant role in determining the availability of sunlight for the plants, in addition to crop selection. The spacing between shelves, the height of the shelves, and the arrangement of the plants can influence the penetration of sunlight and the amount of shading experienced by the lower levels. By carefully considering these design factors, it is possible to improve light distribution and promote more uniform growth and productivity across the vertical farming system (Choubchilangroudi & Zarei, 2022; Lubna et al., 2022).

The comparison between vertical and horizontal farming clearly demonstrates the advantage of vertical farming in terms of crop yield and economic viability (*Touliatos, Dodd & Mcainsh, 2016*; *Beacham, Vickers & Monaghan, 2019*; *Oh & Lu, 2023*). In the same unit area, more crops were planted in vertical farming than horizontal farming (Table 2). The total average fresh yields of vertical farming (6.34 kg m$^{-2}$) were higher than those of horizontal farming (2.11 kg m$^{-2}$), as shown in Table 3. Unlike horizontal farming, vertical farming achieves higher yields by utilizing a greater number of plant containers within the same floor unit space. Moreover, the fresh weight of lettuces planted in vertical farming was much higher than in horizontal farming. This three-fold difference in yield emphasizes the potential of vertical farming to maximize crop production within limited space (*Van Gerrewey, Boon & Geelen, 2022*; *Zhu & Marcelis, 2023*).

Furthermore, the economic benefits of vertical farming are evident from the total income comparison. The total income from vertical farming was higher than that from horizontal farming, with values of 14.5 and 4.8 USD, respectively (Table 3). These findings underscore the financial advantages of vertical farming, which can provide higher returns on investment due to increased crop yields and productivity. However, the result of the financial analysis showed that the investment cost of vertical farming on shelve construction was three times greater than that of horizontal farming. At the same time, triple amounts of irrigated water were observed from vertical farming compared to horizontal farming, as pointed out by the study of *Wicharuck et al. (2023)*. Therefore, further study regarding different crop types growing on the vertical shelves and economic values must be investigated. Each crop may have specific growth requirements and market demands, which must be considered when implementing vertical farming systems. Assessing the feasibility, profitability, and market potential of different crop varieties in vertical farming will contribute to a more comprehensive understanding of the economic benefits and potential challenges associated with this agricultural approach.

Moreover, additional investigations into the optimization of vertical farming techniques, such as light distribution, nutrient management, and environmental control, are essential for maximizing crop growth and achieving even higher yields. Fine-tuning the design and operational parameters of vertical farming systems can further enhance productivity and economic efficiency. Therefore, future research on vertical farming topics should concentrate on developing advanced light management strategies to optimize crop performance in vertical farming. This includes exploring dynamic lighting systems, spectrum manipulation techniques, and light distribution optimization methods to ensure that crops receive adequate and appropriate light levels for optimal growth and productivity. Additionally, further investigations into the responses of different lettuce cultivars to varying light intensities and other environmental factors will help identify well-suited cultivars for vertical farming systems.

## CONCLUSIONS

The research study provides important insights into the potentiality of vertical farming for lettuce production in limited space. The findings reveal that sunlight availability within the vertical farming system diminishes from the upper to lower levels, resulting in reduced

light intensity. Higher light intensity did not always improve crop growth and yields. Crop growth (plant height and canopy width) and yields of both lettuce varieties were observed to be higher on the middle and lower levels compared to the upper level. These observations highlight the significance of crop selection and the optimization of specific crop types for each level based on light intensity considerations. It is crucial to select lettuce cultivars that can thrive under varying light conditions and align with the available light intensity on different levels of the vertical farming system.

## ACKNOWLEDGEMENTS

Special thanks to the (i) Energy Technology for Environment Research Center, Faculty of Engineering, Chiang Mai University and (ii) Faculty of Agriculture, Chiang Mai University, Chiang Mai, Thailand, for the support and laboratory work.

### Funding

This research project was supported by Fundamental Fund 2024, Chiang Mai University. The funders had no role in study design, data collection and analysis, decision to publish, or preparation of the manuscript.

### Grant Disclosures

The following grant information was disclosed by the authors:
Fundamental Fund 2024, Chiang Mai University.

### Competing Interests

The authors declare that they have no competing interests.

### Author Contributions

- Suwimon Wicharuck conceived and designed the experiments, analyzed the data, prepared figures and/or tables, authored or reviewed drafts of the article, and approved the final draft.
- Nuttapon Khongdee conceived and designed the experiments, analyzed the data, prepared figures and/or tables, authored or reviewed drafts of the article, and approved the final draft.
- Ar Man performed the experiments, prepared figures and/or tables, and approved the final draft.
- Wahyu Nurkholis Hadi Syahputra performed the experiments, analyzed the data, prepared figures and/or tables, and approved the final draft.
- Parichat Yalangkan performed the experiments, prepared figures and/or tables, and approved the final draft.
- Prapaporn Chaiphak performed the experiments, prepared figures and/or tables, and approved the final draft.
- Chatchawan Chaichana conceived and designed the experiments, prepared figures and/or tables, authored or reviewed drafts of the article, and approved the final draft.

## Data Availability

The raw data are available in the Supplemental File.

## Supplemental Information

Supplemental information for this article can be found online at http://dx.doi.org/10.7717/peerj.17085#supplemental-information.

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
