# Peer review of "Vertical farming for lettuce production in limited space: a case study in Northern Thailand"

_PeerJ, doi:10.7717/peerj.17085_

## Round 0.1 · original submission · Major Revisions

Three reviewers have now commented, and as you can see Reviewers 1 and 2 have substantial comments which need to be addressed.

Please address each of the reviewers' comments individually.

Reviewer 1 ·

Basic reporting

The vertical farming used in this study is a spatially friendly application, and in recent years, there has also been increasing research on vertical farming. When evaluated in this respect, the article has an original value. However, the English used in this study should be checked carefully and improved. Besides that, Sufficient research background has not been provided. The chart for this study is acceptable and the raw data was shared.

Experimental design

The content of this study is suitable for this journal but this study did not provide sufficient research background and the research purpose was not very clear. This study did not specify the specific experimental design in the main text, and the description of the experimental method requires more details.

Validity of the findings

This study provides some data and provides strong support for the results of this study.

Additional comments

1. Please check the expression in lines 43-44, it does not match your results.
2. Please keep the full name of PPFD consistent throughout the text.
3. The background of lettuce in this field should be emphasized, especially in previous studies on the impact of light on lettuce. Why is lettuce chosen as an experimental plant? Just because the growth cycle is short? Line 94.
4. What is the research gap? If this study wants to explore the vertical farms of lettuce in Thailand, the introduction section should focus on the current situation and existing problems of vertical farms in Thailand, as well as the cultivation and consumption of lettuce in Thailand. Then, through this research, further explore or solve the existing problems. Lines 107-110.
5. Regardless of the amount of growing media, does 50 grams remain unchanged? Is charcoal and rock phosphate 50 grams in total or 50 grams each? What is the proportion if there is a total of 50 grams? Line 140.
6. What does "14 days after seedling" mean? Should it be seeding or sowing? What is the date of your sowing? Line 144.
7. What is your experimental design?
8. Some results are not consistent with the results shown in the figures. Lines 214 and 251.
9. From this study's results, the effects of shades are positive. But here is the opposite discussion. Line 298.
10. Please carefully check the format and order of the references. Lines 412, 414, and 437.

Annotated reviews are not available for download in order to protect the identity of reviewers who chose to remain anonymous.

Reviewer 2 ·

Basic reporting

Overall writing is comprehensible but needs some improving. A few citation issues needed to be addressed (see my additional comment).
Table 1 is really nor necessary.

Experimental design

I think the overall experimental design is ok, more details are needed in the Materials and Methods section. It would be better if this study was repeated again and use time as rep.

Validity of the findings

Discussion section is missing morphological effect of different light intensity from upper, middle and lower levels. Currently Discussion section only focused on physiological aspect.

Additional comments

Line 43-44: This sentence sounds a bit awkward, can change to 'Light use efficiency (LUE) decreased trend from the upper to lower levels in both varieties..."
Line 118-119: please add standard deviation to environmental condition values.
Line 143-144: please specify the transplant production condition or source of lettuce transplants.
Line 145: 'after seedling'? After seeding or after sowing.
Line 153-154: Where is the absolute growth rate?
Line 137-174: did you do any repeats? Did you do ANOVA with each shelf as a rep? If you measured FW and DW of individual plant then that's not true reps and your ANOVA would not be statistically sound.
Line 168-174: Please specify the statistical software used.
Line 166: I guess cumulative received sunlight was calculated as sum of DLI for the growing cycle (28 days), need a sentence or an equation explaining how this was calculated.
Line 170: (height and canopy width)
Line 180 -185: I understand that you trying to express with the statement that " Measured PPFD for each level showed the same trends during the measurement periods..." but I politely disagree. It is clear that the middle and lower shelves were shaded by the top shelves around solar noon but top shelves were not strongly shaded by other greenhouse structures. I recommend modifying this statement, maybe to something like, "Measured PPFD for each level showed pattern of sun movement across the sky..."
Line 190 -191: This sentence is confusing, maybe "The middle and lower levels received 32.7% and 53.8% less sunlight compared to the upper level, respectively"
Line 194 -196: also confusing
Line 196 -197: please provide standard deviation. The accumulated DLI was referred to as 'cumulative received sunlight', please chose one term and be consistent.
Line 198-202: a very nice elaboration of figure 2, but I would recommend moving it to the Discussion section
Line 232-237: I suggest moving to discussion section
Line 253: don't need citation
Line 255-258: I suggest you delete these sentences.
Line 270-271: I suggest changing to "as determined by the Tukey HSD test."
Line 283: correct reference format
Line 290 - 291: delete lower or reduced
Line 292: sunlight intensity is more than twice that at lower levels.
Line 295 - 296: shading obviously affect light intensity...
Line 296-300: Change this sentence. The paragraph emphasized that high PPFD at upper level did not result in highest yield, but this concluding sentence contradicts this whole paragraph.
Line 304-306: I see what you are trying express with this sentence but this is not right. May be "Increase sunlight intensity will continue to increase crop yield if they do not exceed a threshold...". Also, PPFD in your greenhouse, even at the upper level, probably did not reach light saturation point of lettuce. Lettuce yield stops increase with increasing PPFD before light saturation point is reached. Line 304-310 need edit.
Line 306-307: there are a lot papers you can cite for this, but your citation here is not appropriate, just search for light response curve of lettuce. But I don't think healthy lettuce would have light saturation point below 500.
Line 310-312: need some discussion about different crops and their response to light intensity before leading to this statement. You only has discussion about lettuce yield stopped increasing with high light intensity but lack discussion of yield of other crop may continue to increase under high light.
Line 274-318: Your whole Discussion section discussed light intensity and photosynthesis, which is very important. But in your study, it is also evident that high light intensity inhibited canopy expansion possible through photoreceptor cryptochrome signaling pathway. Low yield on the upper level is also a result of morphological effect induced by high light intensity, but not just high light stress on photosynthesis.
Line 339 - 343: you need to cite your previous study here instead of Carotti et al., 2023.
Line 335 - 352: I like your discussion section, however, much of it was based on the financial analysis from your previous study. I suggest that you tie the discussion section closer to the results from this study. Maybe you can assume that lettuce plants in horizontal farming would receive similar PPFD as your upper level in VF, and do some calculation similar to Table 4. So you won't be discussing results from another study instead of this study.
Line 373-379: I suggest moving this paragraph to Discussion but not in Conclusion.

Caption of figure 3 is wrong in the submission system.
Figure 3, e and f: too many significant digits than necessary in the linear equations and R square values. Same with the text at line 225-230.
Table 3: Explain LH and FW, HP should be HS in the footnote. Also correct reference format in the table caption.

Reviewer 3 ·

Basic reporting

The author uses reliable citing resources, sufficient field background/context provided. The authors provided professional article structure, figures, tables, and have shared the Raw data. The results and conclusion are convincing.

Experimental design

The paper is questions-driven and experimental design, replication, and statistics are convincing.

Validity of the findings

The authors have reliable replication and provide underlying data in the table/figures. The ANOVA and post-test are the correct methods to interpret the data. The results are reliable and the conclusion is promising.

Additional comments

In this manuscript, Suwimon Wicharuck et al conducted vertical farming research to evaluate the effect of sunlight on two representative lettuce varieties (Baby Cos and Green Oak) in Hang Dong district, Chiang Mai province, Thailand. The authors first emphasize the importance of vertical farming in maintaining productivity and economics. To evaluate the sunlight effect in vertical farming, the authors use 6 shelves with 3 levels (upper, middle and lower) to evaluate the lettuce phenotypes including plant height, canopy width, fresh/dry weight, light use efficiency under the recorded sunlight parameters like photosynthetic photon flux density/daily light integral (DLI) using ANOVA and post-hoc test. Collectively, they find the middle and lower levels have potential higher yield compared to upper levels. Overall, the authors used appropriate experimental design and statistics to conduct step-by-step research on sunlight response in the confined environments like vertical farming greenhouse, which is a great case for greenhouse horticulture practices. However, I hope authors can clarify a couple of questions before publication.




Major issues:
In the method section Line 117-179: why do authors use February to study the effect of sunlights in lettuce? Lettuce is cool-season vegetables and can also grow fall and winter in the greenhouse. Is there any specific reason to use spring? Hope authors can clarify a little bit.



Minor Issues:
N/A

---

## Round 0.2 · Minor Revisions

Thanks for your contribution to the improving of this manuscript. Please revise according to Reviewer 1’s comments. Happy New Year!

Reviewer 1 ·

Basic reporting

No comment

Experimental design

No comment

Validity of the findings

No comment

Additional comments

1. Lines 43-44. Light use efficiency (LUE) values showed a decreased trend from the upper to lower levels in both varieties. Please check this sentence, it should be an increased trend.
2. The total number of lettuces per level of each shelf was 25 plants and the total per shelf was 75 plants (25 plants × 3 levels), so there are a total of six shelves, 450 plants. Is there one variety or two varieties on each shelf? If your experimental design is a randomized complete block design, there are two varieties on a shelf, which is six blocks, and one variety on a shelf is three blocks. Please clearly state this issue in the manuscript.
3. Lines 275-277. The characteristics of the vertical shelves (VS) in terms of i) shape of the VS, ii) distance between levels, and iii) height of the VS had a significant role in the incident of light on the plant leaves. Because all shelves are the same in this study, it is inappropriate to say that the shape of the VS, the distance between levels, and the height of the VS had a significant role are significant factors, I suggest you delete this sentence, or if you want to discuss this issue, then you need to cite other studies here.

Reviewer 3 ·

Basic reporting

no comment

Experimental design

no comment

Validity of the findings

no comment

Additional comments

i have no additional comments and I am generally satisfied with the response from the authors.

---

## Round 0.3 · accepted · Accept

Thanks for authors' contribution on the improvement of the manuscript. I confirm the the authors have addressed all the reviewers' comments. I believe now it's ready for publication.

Reviewer 1 ·

Basic reporting

No comment

Experimental design

No comment

Validity of the findings

No comment

Additional comments

This manuscript has been greatly improved by the author's efforts.

Reviewer 2 ·

Basic reporting

Good

Experimental design

Ok

Validity of the findings

OK

Additional comments

I suggested additional discussion points which the authors did not follow. But that's ok, this paper is more applied science than basic plant physiology.